# Association between deep learning–based atrial fibrillation burden and in-hospital mortality

Yongseop Lee[1☯], Yujee Chang[2☯], Jihoon Seo[2], Jung Ah Lee[1], Jung Ho Kim[1], Jin Young Ahn[1], Su Jin Jeong[1], Jun Yong Choi[1], Joon-Sup Yeom[1], Nam Su Ku[1]*, Dukyong Yoon[2,3,4]*

**1** Division of Infectious Diseases, Department of Internal Medicine and AIDS Research Institute, Yonsei University College of Medicine, Seoul, Republic of Korea, **2** Department of Biomedical Systems Informatics, Yonsei University College of Medicine, Seoul, Republic of Korea, **3** Institute for Innovation in Digital Healthcare (IIDH), Severance Hospital, Seoul, Republic of Korea, **4** Center for Digital Health, Yongin Severance Hospital, Yonsei University Health System, Yongin, Republic of Korea

☯ These authors contributed equally to this work.
* dukyong@yuhs.ac (DY); smileboy9@yuhs.ac (NSK)

## Abstract

Despite its clinical significance, research on atrial fibrillation (AF) burden as a dynamic, real-time predictor of adverse outcomes in patients with critical illness is lacking. This study examined the association between high AF burden and in-hospital mortality in critically ill patients, using intensive care unit (ICU) data from the Medical Information Mart for Intensive Care III (MIMIC-III; 2001–2012) and Yongin Severance Hospital (2021–2023). Electrocardiogram waveform data were analyzed using deep learning models to calculate AF burden. Adult ICU patients were included, with exclusion of those aged ≥90 years and those with an AF burden >0.9. AF burden was defined as the ratio of AF waveforms to total waveforms during ICU admission, with a high burden defined as ≥7.0%. Logistic regression and machine learning models were employed to assess the association between AF burden and in-hospital mortality, as well as to evaluate the contribution of AF burden to mortality prediction. From the MIMIC-III database, 7,734 patients were included: 5,734 (74.1%) had a low AF burden (median, 0.3%) and 2,000 (25.9%) had a high AF burden (median, 22.5%). High AF burden was associated with significantly higher in-hospital mortality (18.1% vs. 8.6%, $P<0.001$) and was identified as an independent risk factor (adjusted odds ratio, 1.63; 95% confidence interval, 1.36–1.95; $P<0.001$). Machine learning models demonstrated that AF burden is a significant contributor to mortality prediction, with an area under the curve of 0.86. AF burden may serve as a dynamic marker for real-time alerts of clinical deterioration and for risk stratification in critically ill patients.

**Data availability statement:** Yongin Severance Hospital dataset (external dataset) cannot be shared publicly because they contain potentially identifying and sensitive patient information and are subject to the Personal Information Protection Act and institutional review board (IRB) restrictions. Requests for access to the Yongin Severance Hospital dataset may be directed to the Institutional Review Board of Yonsei University Health System (email: irb@ yuhs.ac). MIMIC-III dataset is publicly available via PhysioNet (MIMIC-III Waveform Database, https://physionet.org/content/mimic3wdb/1.0/), subject to completion of the required data use training and credentialing process. The code used for model development and analysis is publicly available at https://github.com/ CMI-Laboratory/AF_burden.

**Funding:** This work was supported by the National Research Foundation of Korea (NRF) grant funded by the Ministry of Science and ICT of the Republic of Korea (RS-2023-00276320 to D.Y). The funders had no role in study design, data collection and analysis, decision to publish, or preparation of the manuscript.

**Competing interests:** The authors have declared that no competing interests exist.

## Author summary

When people become critically ill and are admitted to the intensive care unit, irregular heart rhythms such as atrial fibrillation are common and can be dangerous. In the past, most research has looked at atrial fibrillation simply as present or absent. However, this approach ignores how much time a patient actually spends in this rhythm. In our study, we measured the total "burden" of atrial fibrillation, meaning the percentage of time a patient's heart was in this rhythm during their stay in the intensive care unit. We analyzed over 7,700 patients from a large public hospital database in the United States and confirmed our results using data from a Korean hospital. We found that patients with a high atrial fibrillation burden had a significantly higher risk of dying in the hospital. Using deep learning and machine learning methods, we also showed that atrial fibrillation burden was an important factor in predicting patient outcomes, alongside age, sepsis, and use of a ventilator. Because heart rhythm monitoring is already part of routine care in intensive care units, our approach could allow doctors to identify high-risk patients in real time, without extra cost or procedures, and potentially guide early interventions.

## Introduction

Atrial fibrillation (AF) is a common form of arrhythmia diagnosed in critically ill patients. The incidence of new-onset AF in patients with critical illnesses ranges from 4.5–15.0%. The risk increases with advanced age, acute respiratory failure, and sepsis, particularly in patients with septic shock, where the incidence increases to 46% [1,2]. The development of AF increases the risk of in-hospital mortality and a prolonged length of stay [2,3].

Most studies have evaluated AF as a binary entity (present or absent). However, the AF burden (the total amount of time spent in AF rhythm) has recently been suggested as a crucial predictor of AF-related adverse outcomes as opposed to the presence of AF [4]. Importantly, AF burden represents a dynamic marker that can only be accurately captured through continuous, real-time monitoring rather than intermittent ECG assessments. Prior studies relying on spot recordings or manual review were unable to reflect temporal fluctuations in AF episodes.

Previously, technology for processing large amounts of electrocardiogram (ECG) data was lacking. Recent advances in deep learning have enabled the automatic classification of heart rhythms at the cardiologist level, facilitating accurate and effective analysis of large volumes of ECG monitoring data [5]. Leveraging these advances, deep learning models now enable continuous and automatic computation of AF burden from full-length ICU ECG streams, offering a unique approach compared with conventional intermittent monitoring methods. However, this capability has not been explored in critically ill populations.

In parallel, deep learning has also shown strong performance in other continuous biosignal and IoT-based monitoring environments, where sensor-fused data streams

are analyzed to detect clinically meaningful physiologic patterns [6]. These developments highlight the broader techno-logical foundation supporting our approach and further emphasize the potential of AI-based, real-time signal analysis for improving clinical risk stratification.

Despite the clinical importance of AF in patients with critical illness, research on the AF burden in this population is lacking. Traditionally, scoring systems, such as the Acute Physiology and Chronic Health Evaluation II (APACHE II), Sequential Organ Failure Assessment (SOFA), and Simplified Acute Physiology Score (SAPS), have been widely used to predict the prognosis of patients in the intensive care unit (ICU) [7,8]. However, these methods are limited by their inability to reflect real-time changes in a patient's condition.

This study aimed to examine the association between a high AF burden and in-hospital mortality in patients with critical illness. Additionally, the study sought to determine the contribution of AF burden to in-hospital mortality using machine learning. We also aimed to determine the utility of real-time ECG data in predicting patient mortality in the ICU. An over-view of this study is provided in Fig 1.

## Results

### Study population

From the MIMIC-III database, 57,786 patients were admitted to the ICU, of which 8,455 had ECG data available for analy-sis (Fig 2A). Among these, 721 patients were excluded due to age or suspected persistent AF: 16 patients were aged <18 years, 332 were aged >90 years, and 373 had an AF burden of >0.9. After exclusion, 7,734 patients were included in the analysis; 5,734 (74.1%) and 2,000 (25.9%) had low and high AF burden, respectively.

The median age of the high AF burden group was 74 years (IQR: 64–81 years), which was significantly higher than that of the low AF burden group (60 years [IQR: 49–71 years]; Table 1). The sex distribution was not significantly differ-ent between the two groups, with the proportion of male patients being 57.0% and 58.8% in the low and high AF burden groups, respectively.

The median AF burden for all patients was 1.0% (IQR: 0.1%–7.6%). The median AF burden was 0.3% (IQR: 0.0%–1.5%) and 22.5% (IQR: 12.8%–44.8%) in the low and high AF burden groups, respectively. Significant differences in demographics, admission type, comorbidities, and severity indices were observed between the two groups. The high AF burden group had a higher Charlson comorbidity index, SOFA score, and proportion of patients with sepsis than did the low AF burden group. The characteristics of the study population are presented in Table 1.

### Outcomes

In the MIMIC-III database, the in-hospital mortality rate was significantly higher in patients with a high AF burden (18.1%) than in those with a low AF burden (8.6%; $P<0.001$; Table 2). In-hospital mortality rates increased sequentially with AF burden, ranging from 8.0% in the lowest-burden group to 20.3% in the highest-burden group (S1 Fig). Overall mortality was significantly higher in the high AF burden group (48.4%) than in the low AF burden group (30.0%; $P<0.001$).

Additionally, patients with a high AF burden had a significantly longer median ICU stay (3.0 days [IQR: 1.5–6.0 days]) than did the low AF burden counterparts (2.1 days [IQR: 1.2–4.3 days]; $P<0.001$). The median length of hospital stay was also significantly longer in the high AF burden group (8.0 days [IQR: 4.0–13.0 days]) than in the low AF burden group (7.0 days [IQR: 4.0–12.0 days]; $P<0.001$).

### Impact of AF burden on in-hospital mortality

Logistic regression analysis was conducted to evaluate the association between the AF burden and in-hospital mortality (Table 3). Univariate analysis showed a significant association between high AF burden and in-hospital mortality (OR, 2.35; 95% CI, 2.03–2.72; $P<0.001$). Multivariable analysis was performed to adjust for confounders, including age, sex,

## Study Population

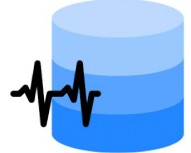

**MIMIC-III database**
ECG waveform
n = 7,734

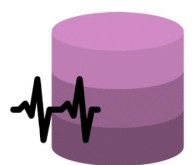

External validation dataset
**Yongin Severance Hospital**
ECG waveform
n = 3,428

## AF burden Calculation

① 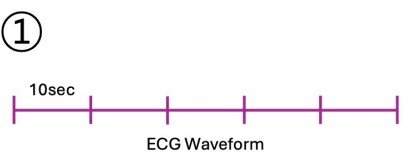

Divide into 10-s segments

② 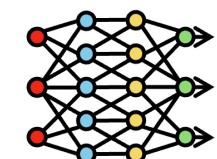

Classify AF rhythm by
using SE-ResNet-34

③ 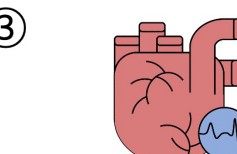

- AF Burden ≥ 7%
    → High AF Burden Group
- AF Burden < 7%
    → Low AF Burden Group

**AF burden**: ratio of AF waveforms to the total number of waveforms during ICU admissions

## Outcome: In-hospital Mortality

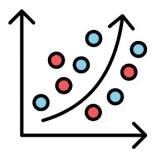

- Conducted logistic regression to evaluate the association between AF burden and in-hospital mortality (Univariable and Multivariable analyses)
- Confounders: Age, Sex, Admission type, CCI, SOFA score, sepsis

## In-hospital mortality Prediction with ML

Random Forest
Train:Test = 8:2

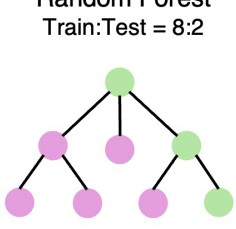

External validation

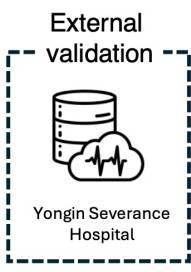

Yongin Severance Hospital

Performance metrics

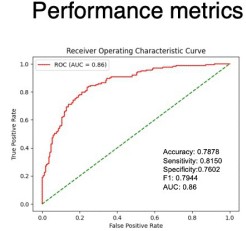

SHAP analysis

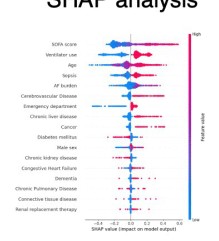

- Specific task: Determine the contribution of AF burden to in-hospital mortality
- Features: AF burden, Age, Sex, Admission type, Comorbidities
- Outcome Interpretation: High AF burden increases the risk of in-hospital mortality

**Fig 1. Study overview. Abbreviations:** MIMIC-III, Medical Information Mart for Intensive Care III; AF, atrial fibrillation; ICU, intensive care unit; CCI, Charlson Comorbidity index; SOFA, Sequential Organ Failure Assessment ML, Machine Learning; SHAP, SHapley Additive exPlanation.

(A)

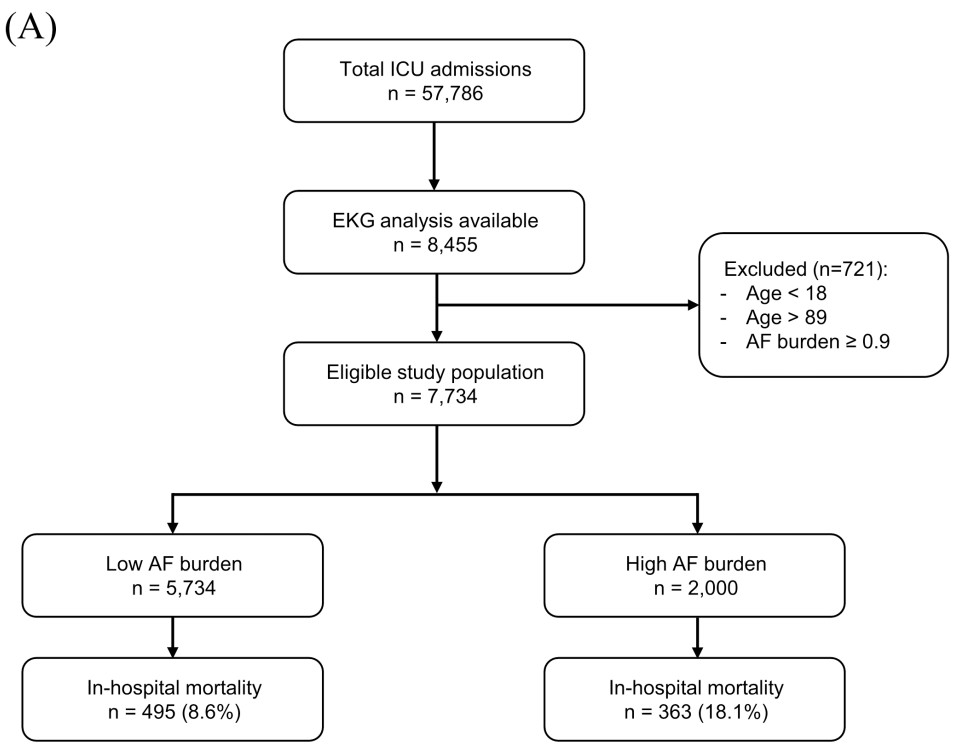

(B)

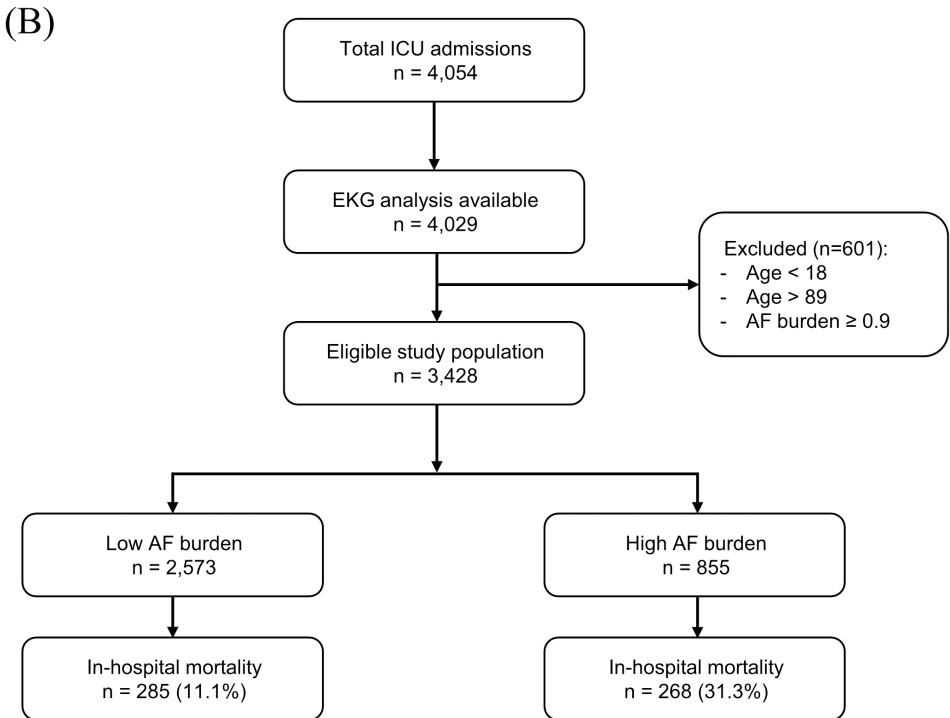

**Fig 2. Flowchart of the study. (A)** MIMIC-III (training dataset) and **(B)** Yongin Severance dataset (external validation dataset) flowcharts. **Abbreviations**: ICU, intensive care unit; EKG, electrocardiogram; AF, atrial fibrillation; MIMIC-III, Medical Information Mart for Intensive Care III.

**Table 1. Clinical characteristics of patients with critical illness stratified by atrial fibrillation burden.**

| | MIMIC-III | | | | Yongin Severance Hospital | | | |
|---|---|---|---|---|---|---|---|---|
| | Total (n=7,734) | Low AF burden (n=5,734) | High AF burden (n=2,000) | P-value | Total (n=3,428) | Low AF burden (n=2,573) | High AF burden (n=855) | P-value |
| Age (years) | 64.0 (52.0–75.0) | 60.0 (49.0–71.0) | 74.0 (64.0–81.0) | <0.001 | 68.0 (57.0–79.0) | 65.0 (54.0–76.0) | 78.0 (67.0–83.0) | <0.001 |
| Male sex | 4,444 (57.5%) | 3,268 (57.0%) | 1,176 (58.8%) | 0.159 | 2,087 (60.8%) | 1,804 (70.1%) | 502 (58.7%) | 0.188 |
| **Admission type** | | | | | | | | |
| Elective admission | 1,191 (15.4%) | 875 (15.3%) | 316 (15.8%) | 0.564 | 906 (26.4%) | 769 (29.9%) | 137 (16.0%) | <0.001 |
| Emergency department | 6,543 (84.6%) | 4,859 (84.7%) | 1,684 (84.2%) | 0.564 | 2,522 (73.6%) | 1,804 (70.1%) | 718 (84.0%) | <0.001 |
| **Admission unit** | | | | | | | | |
| MICU | 2,232 (28.9%) | 1,684 (29.4%) | 548 (27.4%) | 0.094 | 2,137 (62.3%) | 1,531 (59.5%) | 606 (70.9%) | <0.001 |
| SICU | 1,823 (23.6%) | 1,459 (25.4%) | 364 (18.2%) | <0.001 | 1,291 (37.7%) | 1,042 (40.5%) | 249 (29.1%) | <0.001 |
| CCU | 1,680 (21.7%) | 1,152 (20.1%) | 528 (26.4%) | <0.001 | – | – | – | – |
| CSRU | 1,308 (16.9%) | 873 (15.2%) | 435 (21.8%) | <0.001 | – | – | – | – |
| TSICU | 691 (8.9%) | 566 (9.9%) | 125 (6.2%) | <0.001 | – | – | – | – |
| AF burden (%) | 1.0 (0.1–7.6) | 0.3 (0.0–1.5) | 22.5 (12.8–44.8) | <0.001 | 1.1 (0.1–7.0) | 0.4 (0.1–1.7) | 23.6 (12.6–46.8) | <0.001 |
| **Comorbidities** | | | | | | | | |
| Diabetes mellitus | 2,202 (28.5%) | 1,595 (27.8%) | 607 (30.4%) | 0.031 | 2,024 (59.0%) | 1,475 (57.3%) | 549 (64.2%) | <0.001 |
| Congestive heart failure | 2,132 (27.6%) | 1,306 (22.8%) | 826 (41.3%) | <0.001 | 779 (22.7%) | 496 (19.3%) | 283 (33.1%) | <0.001 |
| Chronic kidney disease | 1,300 (16.8%) | 868 (15.1%) | 432 (21.6%) | <0.001 | 521 (15.2%) | 314 (12.2%) | 207 (24.2%) | <0.001 |
| Chronic liver disease | 992 (12.8%) | 787 (13.7%) | 205 (10.2%) | <0.001 | 286 (8.3%) | 227 (8.8%) | 59 (6.9%) | 0.078 |
| Chronic pulmonary disease | 1,532 (19.8%) | 1,085 (18.9%) | 447 (22.4%) | <0.001 | 459 (13.4%) | 301 (11.7%) | 158 (18.5%) | <0.001 |
| Cerebrovascular disease | 1,130 (14.6%) | 827 (14.4%) | 303 (15.2%) | 0.428 | 780 (22.8%) | 601 (23.4%) | 179 (20.9%) | 0.143 |
| Dementia | 401 (5.2%) | 253 (4.4%) | 148 (7.4%) | <0.001 | 52 (1.5%) | 29 (1.1%) | 23 (2.7%) | 0.001 |
| Connective tissue disease | 256 (3.3%) | 170 (3.0%) | 86 (4.3%) | 0.004 | 41 (1.2%) | 33 (1.3%) | 8 (0.9%) | 0.419 |
| Cancer | 547 (7.1%) | 430 (7.5%) | 117 (5.8%) | 0.013 | 414 (12.1%) | 322 (12.5%) | 92 (10.8%) | 0.173 |
| Charlson comorbidity index | 2.0 (1.0–4.0) | 2.0 (1.0–4.0) | 2.0 (1.0–4.0) | <0.001 | 6.0 (3.0–10.0) | 5.0 (3.0–9.0) | 8.0 (5.0–12.0) | <0.001 |
| SOFA score | 3.0 (2.0–6.0) | 3.0 (2.0–5.0) | 4.0 (2.0–7.0) | <0.001 | 4.0 (2.0–8.0) | 3.0 (1.0–7.0) | 7.0 (4.0–11.0) | <0.001 |
| Ventilator use | 3,905 (50.5%) | 2,812 (49.0%) | 1,093 (54.6%) | <0.001 | 935 (27.3%) | 596 (23.2%) | 339 (39.6%) | <0.001 |
| Renal replacement therapy | 368 (4.8%) | 249 (4.3%) | 119 (6.0%) | 0.004 | 395 (11.5%) | 202 (7.9%) | 193 (22.6%) | <0.001 |
| Vasopressor use | 1279 (16.7%) | 813 (14.4%) | 466 (23.6%) | <0.001 | – | – | – | – |
| Sepsis | 2,379 (30.8%) | 1,685 (29.4%) | 694 (34.7%) | <0.001 | 883 (25.8%) | 542 (21.1%) | 341 (39.9%) | <0.001 |

AF, atrial fibrillation; MICU, medical intensive care unit; SICU, surgical intensive care unit; CCU, coronary care unit; CSRU, cardiac surgery recovery unit; TSICU, trauma surgical intensive care unit; SOFA, Sequential Organ Failure Assessment; MIMIC-III, Medical Information Mart for Intensive Care III.

admission type, admission unit, Charlson Comorbidity Index, SOFA score, ventilator use, renal replacement therapy, vasopressor use, and sepsis. After adjustment, a high AF burden remained an independent risk factor for in-hospital mortality in patients with critical illness (adjusted OR, 1.63; 95% CI, 1.36–1.95; P<0.001). Furthermore, restricted cubic spline analysis demonstrated a significant association between AF burden and in-hospital mortality (P<0.001), with significant nonlinearity in this relationship (P<0.001; S2 Fig).

**Table 2. Comparison of outcomes in patients with critical illness stratified by atrial fibrillation burden.**

| | MIMIC-III | | | Yongin Severance Hospital | | |
|---|---|---|---|---|---|---|
| | Low AF burden (n = 5,734) | High AF burden (n = 2,000) | *P*-value | Low AF burden (n = 2,573) | High AF burden (n = 855) | *P*-value |
| In-hospital mortality | 495 (8.6%) | 363 (18.1%) | <0.001 | 285 (11.1%) | 268 (31.3%) | <0.001 |
| Overall mortality | 1721 (30.0%) | 967 (48.4%) | <0.001 | 372 (14.5%) | 319 (37.3%) | <0.001 |
| Length of ICU stay (days) | 2.1 (1.2–4.3) | 3.0 (1.5–6.0) | <0.001 | 2.0 (2.0–4.0) | 3.0 (2.0–6.0) | <0.001 |
| Length of hospital stay (days) | 7.0 (4.0–12.0) | 8.0 (4.0–13.0) | <0.001 | 12.0 (5.0–12.0) | 15.0 (7.0–29.0) | <0.001 |

AF, atrial fibrillation; ICU, intensive care unit; MIMIC-III, Medical Information Mart for Intensive Care III.

**Table 3. Risk factors for in-hospital mortality in patients with critical illness.**

| | Univariable analysis | | Multivariable analysis | |
|---|---|---|---|---|
| | OR (95% CI) | *P*-value | aOR (95% CI) | *P*-value |
| Age | 1.03 (1.03–1.04) | <0.001 | 1.03 (1.03–1.04) | <0.001 |
| Male sex | 0.92 (0.80–1.06) | 0.241 | 0.90 (0.77–1.06) | 0.220 |
| MICU | 1.75 (1.51–2.02) | <0.001 | 1.10 (0.92–1.31) | 0.305 |
| SICU | 0.96 (0.81–1.13) | 0.595 | | |
| CCU | 0.96 (0.80–1.14) | 0.637 | | |
| CSRU | 0.35 (0.26–0.45) | <0.001 | 0.23 (0.17–0.32) | <0.001 |
| TICU | 1.02 (0.79–1.30) | 0.865 | | |
| Admission via ED | 4.23 (3.09–5.96) | <0.001 | 2.54 (1.79–3.71) | <0.001 |
| Charlson Comorbidity Index | 1.18 (1.15–1.21) | <0.001 | 1.09 (1.06–1.13) | <0.001 |
| High AF burden | 2.35 (2.03–2.72) | <0.001 | 1.63 (1.36–1.95) | <0.001 |
| SOFA score | 1.30 (1.27–1.33) | <0.001 | 1.21 (1.18–1.25) | <0.001 |
| Ventilator use | 4.20 (3.55–4.99) | <0.001 | 3.90 (3.20–4.76) | <0.001 |
| Renal replacement therapy | 2.39 (1.84–3.08) | <0.001 | 0.92 (0.67–1.27) | 0.629 |
| Vasopressor use | 3.85 (3.30–4.49) | <0.001 | 1.06 (0.86–1.32) | 0.575 |
| Sepsis | 3.71 (3.21–4.30) | <0.001 | 1.36 (1.14–1.62) | <0.001 |

All variance inflation factors were <5, and the Hosmer–Lemeshow test indicated no significant lack of fit (*P* = 0.203).

OR, odds ratio; aOR, adjusted odds ratio; AF, atrial fibrillation; MICU, medical intensive care unit; SICU, surgical intensive care unit; CCU, coronary care unit; CSRU, cardiac surgery recovery unit; TSICU, trauma surgical intensive care unit; ED, emergency department; SOFA, Sequential Organ Failure Assessment.

## Prediction of in-hospital mortality using machine learning models based on AF burden

Machine learning models were trained to predict in-hospital mortality, and the SHAP values were measured. Various models, including the Random Forest Classifier, XGBoost, and Logistic Regression, were trained, with the Random Forest Classifier model achieving the best performance (S1 Table). The performance metrics of the Random Forest Classifier model included an accuracy of 0.788, sensitivity of 0.815, specificity of 0.760, positive predictive value (PPV) of 0.775, F1 score of 0.795, and area under the curve (AUC) of 0.86 (Fig 3A).

Using the SHAP value statistical analysis, the AF burden had an impact on the prediction of in-hospital mortality (Figs 3C and S3A). The AF burden was ranked after SOFA score, ventilator use, age, and sepsis in terms of feature importance. This high ranking underscores the significant role of AF burden in the model's predictions. Higher AF burden values were associated with increased mortality, demonstrating the importance of AF burden in effectively predicting patient outcomes.

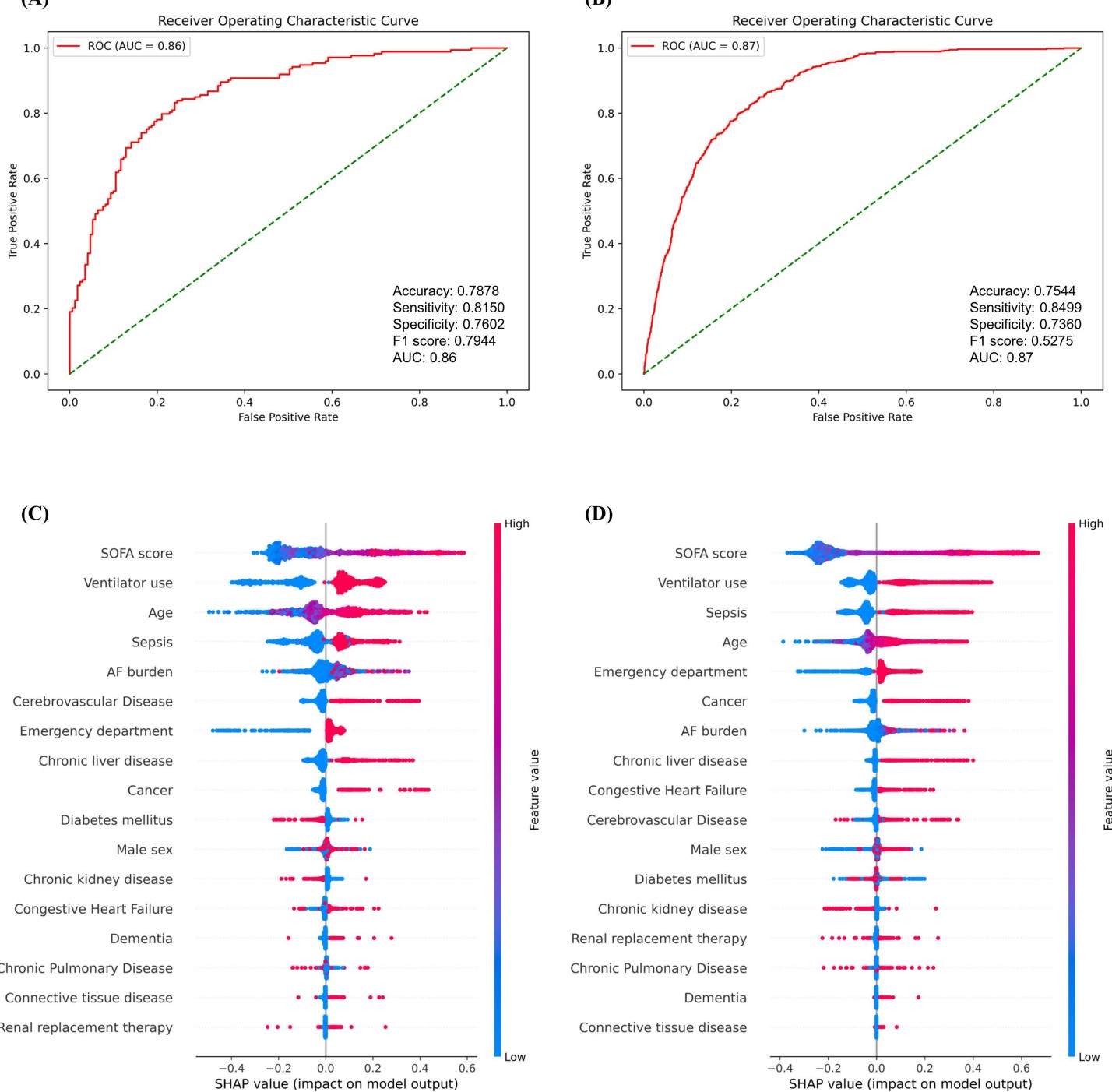

**Fig 3. Performance and feature importance of in-hospital mortality prediction models in patients with critical illness. (A)** MIMIC-III (training dataset) AUROC, **(B)** Yongin Severance Hospital (external validation dataset) AUROC, **(C)** MIMIC-III (training dataset) SHAP values, **(D)** Yongin Severance Hospital (external validation dataset) SHAP values. **Abbreviations**: MIMIC-III, Medical Information Mart for Intensive Care III. MIMIC-III, Medical Information Mart for Intensive Care III; AUROC, Area Under Receiver Operating Characteristic Curve; SHAP, SHapley Additive exPlanation.

## External validation

For the external validation dataset from the Yongin Severance Hospital, 3,428 eligible patients were included in the analysis (Fig 2B). Among them, 2,573 (75.1%) and 855 (24.9%) patients had low and high AF burdens, respectively. The two groups differed significantly in several baseline characteristics, including age, sex, admission type, admission unit, and comorbidities (Table 1). The median AF burden in all patients was 1.1% (IQR: 0.1–7.0). The median AF burden was 0.4% (IQR: 0.1–1.7) and 23.6% (IQR: 12.6–46.8) in the low and high AF burden groups, respectively. The high AF burden group had significantly higher in-hospital (31.3% vs. 11.1%; $P < 0.001$) and overall mortality (37.3% vs. 14.5%; $P < 0.001$) rates than the low AF burden group.

The performance of the machine learning model in predicting in-hospital mortality was evaluated using a Yongin Severance Hospital external validation dataset. The model showed an accuracy of 0.754, sensitivity of 0.850, specificity of 0.736, PPV of 0.382, F1 score of 0.528, and AUC of 0.87 (Fig 3B). These performance metrics are comparable to the results obtained from the MIMIC-III training dataset. The SHAP values also showed that AF burden had a significant impact on the model predictions (Fig 3D and S3B).

## Discussion

We investigated the association between a high AF burden and in-hospital mortality in patients with critical illnesses. Patients with high AF burden had a significantly increased risk of in-hospital mortality. A substantial proportion of patients with critical illness had a high AF burden, with mortality rates exceeding 20% and the highest AF burden (>23.9%). We introduced a method to calculate AF burden using a deep learning model applied to continuously recorded ECG waveform data in the ICU. Furthermore, the findings were validated using external data from Yongin Severance Hospital, demonstrating the robustness and generalizability of the results.

Although AF has long been recognized as a marker of poor prognosis in patients with critical illness, it has primarily been understood as a binary entity [9]. However, recent studies have evaluated the impact of AF burden—the proportion of time spent in AF during monitoring—and have shown that a high AF burden is significantly associated with subsequent AF diagnosis and ischemic stroke [10,11]. In the study by Lancini et al., a threshold of ≥7% was used to define high AF burden, corresponding to the median AF burden of their cohort [12]. The authors noted that this cutoff was used as an operational threshold for binary analyses rather than a clinically validated value. Importantly, they demonstrated that AF burden was a strong independent predictor of subsequent AF diagnosis, with risk increasing progressively across higher burden levels—supporting the interpretation of AF burden as a continuous risk factor.

Similarly, in our study, a high AF burden was significantly associated with in-hospital mortality among patients with critical illness. We also used the 7% threshold applied by Lancini et al [12]. to distinguish high AF burden; however, this value served only as a reference point for binary analysis, as our objective was not to propose a fixed clinical cutoff for high AF burden but to evaluate its continuous association with mortality risk. As shown in S1 and S2 Figs, mortality increased steadily with higher AF burden, illustrating that AF burden behaves as a continuous risk factor. Moreover, restricted cubic spline analysis showed significant nonlinearity between AF burden and mortality. This nonlinear pattern supports the use of flexible approaches, including our deep learning–based mortality prediction model, which can account for complex, non-linear relationships beyond the capability of linear models.

A possible explanation for the observed association between AF burden and mortality is that AF burden may serve as a surrogate marker of underlying illness severity and systemic stress, given its frequent occurrence in the setting of heightened inflammatory responses, vasopressor use, electrolyte disturbances, and advanced organ dysfunction [13]. Additionally, AF burden may contribute to clinical deterioration, as sustained AF can eliminate effective atrial contraction, reduce diastolic filling, and increase ventricular rates, which together can impair cardiac output and worsen hemodynamic stability in vulnerable ICU patients [13]. These mechanisms support the clinical plausibility of the observed association between

higher AF burden and increased mortality. Hence, AF should be understood not as a binary entity, but as a continuous variable known as the AF burden.

Although numerous mortality prediction models have been proposed, most rely exclusively on static clinical variables such as age, sepsis, and SOFA score, which are collected intermittently and do not capture rapid physiologic deterioration [7,8]. A key innovation of our study is the incorporation of a continuously quantified AF burden derived from deep learning analysis of real-time ICU ECG waveforms. By transforming AF from a binary event into a dynamic, time-dependent physiologic marker, our model captures high-resolution cardiac instability that precedes clinical deterioration—an aspect that existing models cannot assess. Importantly, AF burden provided independent and incremental prognostic value beyond traditional predictors, and its significance was consistently reproduced in an external ICU cohort. These findings demonstrate that leveraging continuous monitoring data introduces a novel physiologic dimension to mortality prediction that meaningfully extends beyond the capabilities of conventional risk scores. Building on this advantage, real-time AF burden monitoring could be integrated into ICU clinical decision-support systems to facilitate earlier detection of clinical deterioration. Dynamic AF burden thresholds could trigger automated alerts, prompting clinicians to evaluate hemodynamic status, reassess rate or rhythm-control strategies, and consider anticoagulation when appropriate. Incorporating AF burden into existing ICU dashboards would enable continuous, real-time risk stratification and may help identify patients whose cardiovascular stability is worsening.

In the SHAP analysis, this distinction was clearly reflected in the feature contributions. Higher SOFA scores, mechanical ventilation use, and sepsis were strong positive predictors of mortality, consistent with their roles as established markers of critical illness severity [7,8]. AF burden also emerged as an important contributor, functioning as a dynamic real-time marker rather than a static admission-time variable. Increasing AF burden was associated with a steadily higher predicted mortality risk, indicating that sustained or frequent AF episodes capture ongoing hemodynamic instability during the ICU stay. These results further support the added value of continuous monitoring–derived variables in mortality prediction models.

Deep learning offers a promising approach for automating ECG feature extraction, with convolutional and recurrent neural networks demonstrating robustness in distinguishing AF [14]. Traditional convolutional networks, when built with increased layers, often encounter gradient vanishing or explosion, a challenge ResNet overcomes through skip (residual) connections. These allow the model to learn residual functions and train deeper networks efficiently by maintaining information across layers without degrading the learning process [15]. ResNet-based models have achieved a high AF detection accuracy. Andreotti et al. used ResNet-34 with data augmentation on single-lead ECGs and reported a high F1 score [16]. Guan et al. added hidden attention to ResNet, effectively capturing ECG spatiotemporal features and improving both accuracy and interpretability [17]. In a previous study, SE-ResNet was compared with ResNet as a baseline model for different layer depths (18, 34, 50, 101, and 152), showing that SE-ResNet had better classification performance than ResNet for all slices [18]. Here, SE-ResNet also surpassed ResNet, with SE-ResNet-34 used to train the final AF classification model. Beyond AF detection performance, deep learning has also demonstrated substantial utility in other continuous biosignal and IoT-based monitoring environments, where sensor-fused physiologic streams are analyzed to capture subtle, clinically meaningful patterns [6]. This expanding technological foundation supports the broader methodological rationale of our study and underscores how AI-based waveform analysis can address gaps left by traditional, intermittently measured clinical variables.

Importantly, this deep learning–based approach overcomes a key limitation of traditional AF assessment in the ICU, which often relies on intermittent rhythm checks or manual review of telemetry strips. Such methods cannot capture the rapid, high-frequency fluctuations in cardiac rhythm that occur during critical illness. By automatically analyzing every 10-second ECG segment, our model provides continuous, high-resolution quantification of AF burden that would be impractical to obtain manually, thereby reducing sampling bias and clinician workload in a busy ICU environment.

We present methods for real-time risk stratification and prediction of clinical deterioration using the AF burden. While ICU scoring systems such as SOFA, APACHE, and SAPS are widely used, they rely on the manual input of complex

patient characteristics and include discontinuously measured variables, limiting their ability to adequately reflect real-time deterioration [7,8]. In contrast, our approach uses only existing 24-h monitoring data to detect AF burden, requiring no additional input or cost. A key finding is that the AF burden can be calculated in real time using ECG waveforms to predict the outcomes. As ECG monitoring is routine in the ICU, this method enables continuous, cost-free assessment without added intervention.

Patients with a high AF burden may benefit from early and aggressive management, such as antiarrhythmic medications. However, the efficacy of pharmacological agents in the treatment of AF in patients with critical illnesses varies. Amiodarone (30.0%–95.2%), beta-blockers (31.8%–92.3%), and calcium channel blockers (30.0%–87.1%) have shown mixed success rates, with risks of hypotension, especially in patients with sepsis. Magnesium (55.2%–77.8%) appears safer, with fewer adverse events, particularly as a first-line agent or adjunct to amiodarone. Nonetheless, these drugs have side effects, and no study has conclusively shown a survival benefit [19]. Previous studies have treated AF as a binary condition, without exploring rhythm control strategies in high-risk groups identified by AF burden reclassification. This study highlights the need to recognize AF as a continuous variable rather than a binary variable. Further research is warranted to assess whether additional treatment in patients with high AF burden and mortality risk improves prognosis.

This study had several limitations. The cohort included only patients with critical illnesses in the ICU; therefore, the findings may not be generalizable to general ward patients. The AF burden was calculated as the proportion of total ICU stay, limiting the assessment of its real-time implications. Moreover, we could not determine whether the AF burden–guided interventions improved the outcomes. The study also did not identify the specific reason for ICU admission, although ICU type was included. Baseline cardiac history and echocardiographic findings were not available in our dataset and therefore could not be included in the analysis. Future studies should assess whether targeting the AF burden can improve outcomes and whether real-time clinical decisions based on the AF burden offer prognostic benefits.

In conclusion, high AF burden was significantly associated with in-hospital mortality. We present a method to calculate AF burden by analyzing continuous ECG monitoring in the ICU using a deep learning model. The AF burden can be used to provide real-time alerts for clinical deterioration in the ICU and stratify the risk of patients with critical illness and AF.

## Methods

### Study population

This multicenter case–control study utilized data from the Medical Information Mart for Intensive Care III (MIMIC-III) database (version 2.2) and Yongin Severance Hospital [20,21]. The MIMIC-III database is a publicly available de-identified electronic health record database that encompasses comprehensive clinical information and waveform data of patients admitted to the ICU of the Beth Israel Deaconess Medical Center (BIDMC) in Boston, Massachusetts, between 2001 and 2012. The waveform record contained digitized signals, including ECG, arterial blood pressure, and respiratory signals [22]. The data were randomly collected in an automated manner.

Yongin Severance Hospital, located in Yongin, Republic of Korea, is a 708-bed, university-affiliated hospital. The patients included in this study were admitted to the ICU of this hospital between January 2021 and February 2023. ECG data from the hospital were extracted via the Severance Data Portal, which provided access to single-lead 10-s ECG recordings. These data were collected at a 500-Hz sampling rate using an ECG lead II, with a system capable of real-time monitoring in the ICU. These data were used as an external validation dataset for machine learning models assessing the predictive value of AF burden for in-hospital mortality.

This study included adult patients admitted to the ICU of each hospital. Patients aged ≥90 years were excluded because age ≥ 90 years was anonymized in the MIMIC-III database due to patient de-identification issues. Cases in which the AF burden could not be calculated due to incomplete ECG data were excluded. Furthermore, patients with an AF burden > 0.9 were also excluded. An AF burden above this level is highly suggestive of persistent AF, which is defined as

AF lasting longer than 7 days and not self-terminating [23]. In such cases, AF is continuously present regardless of short-term changes in clinical condition, making it less appropriate to use AF burden as a dynamic predictor of outcomes. Given the lack of formal historical diagnostic information (e.g., past ECG data and physician's diagnosis) to reliably exclude all persistent AF cases, the > 0.9 threshold was employed based on this clinical rationale.

**AF burden**

To calculate the AF burden, AF binary convolutional neural network models were trained using public ECG datasets (PTB-XL [24] and AF 2017 Challenge dataset [25], S1 and S2 Methods). The model architectures used were ResNet-18, ResNET-34, SE-ResNet-18, and SE-ResNet-34. Model performance was evaluated using another public ECG dataset (Shaoxing Hospital ECG dataset [26], S3 Method). The SE-ResNet-34 model [27,28] demonstrated the best overall performance (accuracy: 0.943, sensitivity: 0.937, specificity: 0.945, AUROC: 0.983) and was therefore selected as the final AF detection model for computing AF burden from ICU ECG waveforms. (S2 Table). The SE-ResNet-34 architecture consisted of an initial 1-D convolutional layer (kernel size 7, 64 channels, stride 2), batch normalization, ReLU activation, and a max-pooling layer (kernel size 3, stride 2), followed by four residual stages with 3, 4, 6, and 3 blocks and channel widths of 64, 128, 256, and 512, respectively. In each residual block, two 1-D convolutional layers (kernel size 3) with batch normalization and ReLU were followed by a squeeze-and-excitation (SE) block, which applied global average pooling and two fully connected layers to adaptively reweight channel-wise features. A global average pooling layer over the temporal dimension and a final fully connected layer produced the binary AF prediction. To exclude pacemaker rhythm, another SE-ResNet-34 model was trained for pacemaker rhythm prediction. This model was trained using the PTB-XL and Lobachevsky University Electrocardiography Database (S4 Method) [29]. Model performance was evaluated using the MIT-BIH Arrhythmia Database [30] (S5 and S6 Methods, S3 Table).

The entire waveform data of each patient was divided into 10-s segments. Segments with poor signal quality or pacemaker rhythms were excluded from the analysis. The signal quality was assessed using a high-pass filter to remove the baseline wander, setting the cutoff frequency at 0.5 Hz. A power line filter was employed to eliminate the 50-Hz power line noise [31]. Z-normalization was performed using the mean and standard deviation. After exclusion, an AF classification model was applied to each segment. The AF burden was calculated as the ratio of AF waveforms to the total number of waveforms during ICU admission. For the external validation cohort, the identical preprocessing pipeline and AF classification model used for the MIMIC-III cohort were applied without any modification, including the same filtering procedures, segmentation strategy, normalization method, and the pretrained SE-ResNet-34 classifier. AF burden in the external cohort was computed using the same algorithm, ensuring methodological consistency across cohorts.

**Variables and outcome measures**

The following variables were extracted from the MIMIC-III database and the Severance Clinical Research Analysis Portal for Anonymous (SCRAP-A) Yongin Severance Hospital data: demographic characteristics (age and sex), admission type, ICU admission, comorbidities, SOFA score, ventilation, and renal replacement therapy. The outcome of interest was the in-hospital mortality.

In patients with critical illness, a high AF burden was defined as ≥7.0% during the ICU stay, following the threshold used in a prior study by Lancini et al [12]. This cutoff was applied to categorize participants into high and low AF burden groups. Additionally, the AF burden was categorized into four quartiles: Q1, 0.1%–2.4%; Q2, 2.4%–7.0%; Q3, 7.1%–23.9%; and Q4, 24.7%–100% [12]. Sepsis was determined on the basis of the International Classification of Diseases (ICD) ninth (ICD-9) and tenth (ICD-10) revision codes [32,33]. The Charlson Comorbidity Index was calculated using coding algorithms on the basis of ICD-9-CM and ICD-10 codes [34]. These variables were extracted based on the criteria used during ICU admission.

## Statistical analysis

Continuous variables were compared between groups using independent two-tailed t-tests or the Mann–Whitney U-test, as appropriate, based on the data distribution. The Shapiro–Wilk test was used to assess normality, with normally distributed variables reported as means±standard deviations and non-normally distributed variables as medians with interquartile ranges (IQRs). Categorical variables were analyzed using Pearson's chi-square or Fisher's exact test, as appropriate.

Univariable logistic regression was used to assess the association between a high AF burden and in-hospital mortality. Odds ratios (ORs) and 95% confidence intervals (CIs) were calculated for all variables. Variables that showed a statistically significant association ($P<0.05$) in the univariable analysis were included in the multivariable logistic regression model. Multicollinearity was assessed using a variance inflation factor >10. The goodness of fit of the logistic regression model was assessed using the Hosmer–Lemeshow test. In addition, a restricted cubic spline analysis was performed to examine potential nonlinear associations between AF burden and mortality [35]. Covariates that remained significant in the multivariable logistic regression model were included in the spline model to adjust for confounding. Statistical analyses were performed using R, version 4.4.0 (The R Foundation for Statistical Computing, Vienna, Austria). Random Forest model and SHapley Additive exPlanation (SHAP) value analyses were conducted using Python (version 3.10.13) in a Jupyter notebook. The SQL code used for the data extraction is available at GitHub (https://github.com/MIT-LCP/mimic-code/tree/main/mimic-iii).

## Machine learning modeling

Based on the statistical analyses, machine learning models were developed to further explore the predictive power and future importance of AF burden in determining in-hospital mortality. A Random Forest Classifier was implemented using scikit-learn and Random Forest libraries in Python (version 3.10.13) in a Jupyter notebook. The Random Forest model, an ensemble learning method known for its robustness and ability to handle complex feature interactions while mitigating overfitting, was selected for its suitability for classification tasks (S1 Table). The dataset was divided into training and test sets in an 8:2 ratio. Model performance was assessed using metrics, including accuracy, sensitivity, specificity, F1 score, and area under the receiver operating characteristic curve (AUROC). To assess the importance of AF burden and other features in predicting in-hospital mortality, the SHAP values were calculated. SHAP provides a unified measure of the contribution of the features, highlighting the impact of each variable on the model's predictions [36]. To facilitate the interpretation of how different variables influenced the model's decisions, the resulting SHAP values were visualized using different plots. A list of model features, including their name and definitions is provided in S4 Table to enhance clarity and reproducibility.

For external validation, an independent dataset from Yongin Severance Hospital was used, which was also divided into training and test sets at a ratio of 8:2. To assess the generalizability of the model to external data, performance metrics (accuracy, sensitivity, specificity, F1 score, and AUROC) were recalculated.

## Ethics statement

The MIMIC-III database was approved by the institutional review boards of the Massachusetts Institute of Technology and BIDMC. The protocols for the external validation dataset were approved by the Institutional Review Board of Yongin Severance Hospital (Approval No. 9-2023-0040). The requirement for informed consent was waived by the board owing to the retrospective analysis of anonymized data obtained from routine evaluations of patients undergoing Holter monitoring.

## Supporting information

**S1 Method. PTB-XL data.**
(DOCX)

**S2 Method. AF 2017 Challenge data.**
(DOCX)

**S3 Method. Shaoxing Hospital data.**
(DOCX)

**S4 Method. Lobachevsky University Electrocardiography Database (LUDB).**
(DOCX)

**S5 Method. MIT-BIH Arrhythmia database.**
(DOCX)

**S6 Method. Deep-learning model for rhythm classification.**
(DOCX)

**S1 Table. Performance of in-hospital mortality prediction models in critically ill patients.**
(DOCX)

**S2 Table. Performance of the atrial fibrillation classification model using external validation dataset.**
(DOCX)

**S3 Table. Performance of the pacemaker rhythm classification model using external validation dataset.**
(DOCX)

**S4 Table. Feature name and clinical meaning.**
(DOCX)

**S1 Fig. Risk of in-hospital mortality by AF burden.** A) MIMIC-III training dataset, B) Yongin Severance Hospital external validation dataset.
(DOCX)

**S2 Fig. Restricted cubic spline showing the relationship between AF burden and in-hospital mortality in the MIMIC-III dataset.**
(DOCX)

**S3 Fig. Feature importance (A) MIMIC-III dataset, (B) Yongin Severance Hospital.**
(DOCX)

## Author contributions

**Conceptualization:** Yongseop Lee, Yujee Chang, Jihoon Seo, Dukyong Yoon.

**Data curation:** Yongseop Lee, Yujee Chang, Jihoon Seo.

**Formal analysis:** Yongseop Lee, Yujee Chang.

**Funding acquisition:** Dukyong Yoon.

**Investigation:** Yongseop Lee, Yujee Chang, Jihoon Seo, Jung Ah Lee, Jung Ho Kim, Jin Young Ahn, Su Jin Jeong, Jun Yong Choi, Joon-Sup Yeom.

**Methodology:** Yongseop Lee, Yujee Chang, Jihoon Seo.

**Project administration:** Nam Su Ku, Dukyong Yoon.

**Resources:** Dukyong Yoon.

**Software:** Yongseop Lee, Yujee Chang, Jihoon Seo.

**Supervision:** Nam Su Ku, Dukyong Yoon.

**Validation:** Yongseop Lee, Yujee Chang.

**Visualization:** Yongseop Lee, Yujee Chang.

**Writing – original draft:** Yongseop Lee, Yujee Chang.

**Writing – review & editing:** Jung Ah Lee, Jung Ho Kim, Jin Young Ahn, Su Jin Jeong, Jun Yong Choi, Joon-Sup Yeom, Nam Su Ku, Dukyong Yoon.

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
