## [Decision Letter · Decision Letter 0]

13 Nov 2025

Response to Reviewers
Revised Manuscript with Track Changes
Manuscript
**Journal Requirements:**

1. Please send a completed 'Competing Interests' statement, including any COIs declared by your co-authors. If you have no competing interests to declare, please state "The authors have declared that no competing interests exist". Otherwise please declare all competing interests beginning with the statement "I have read the journal's policy and the authors of this manuscript have the following competing interests:"

2. Your current Financial Disclosure states, “The author(s) received no specific funding for this work.”. However, your funding information on the submission form indicates that you received funding from “Ministry of Science and ICT, South Korea”. Please indicate by return email the full and correct funding information for your study and confirm the order in which funding contributions should appear. Please be sure to indicate whether the funders played any role in the study design, data collection and analysis, decision to publish, or preparation of the manuscript.

**Additional Editor Comments:**

Please include a table describing the feature names and their meanings.

There are already many published models on in-hospital mortality; please explain how this study is different or novel compared to prior work. The top predictors mirror what has been repeatedly shown in pulished works (e.g., sepsis, age, SOFA). This limits the innovative value of your model.

Discuss the clinical significance of the key features highlighted in the SHAP plots; what do these findings imply in a medical context? SHAP and mortality common topics, find some new interpretability methods to interpret the model.

The model architectures are rather basic; consider adding some methodological novelty or advanced comparison.

The image quality in the figures needs improvement (resolution).

Indicate whether you have validated the model using an external dataset.

**Reviewers' Comments:**

**Comments to the Author**

1. Does this manuscript meet PLOS Digital Health’s publication criteria?

Reviewer #1: Yes

Reviewer #2: Yes

2. Has the statistical analysis been performed appropriately and rigorously?

Reviewer #1: Yes

Reviewer #2: Yes

3. Have the authors made all data underlying the findings in their manuscript fully available (please refer to the Data Availability Statement at the start of the manuscript PDF file)?

Reviewer #1: Yes

Reviewer #2: Yes

4. Is the manuscript presented in an intelligible fashion and written in standard English?

Reviewer #1: Yes

Reviewer #2: Yes

Reviewer #1: 1. In the Introduction and abstract the authors need, specify AF burden as a dynamic marker produced through real-time monitoring. In the introduction, clearly state the uniqueness of applying deep learning models to continuous and automatic computation of AF burden in critically ill patients compared to those utilizing intermittent monitoring devices.

2. The use consistent notation between the two data sets. For instance, sometimes MIMIC-III is also referred to as "training dataset" and Yongin Severance Hospital as "external validation dataset", which is right, but occasional use of "training dataset" without declaring MIMIC-III (e.g., Line 880) should be removed for clarity.

3. The Data Availability statement indicates that it is not feasible to share the external validation dataset openly due to the Personal Information Protection Act, understandable. However, in simple words, proclaim MIMIC-III as the primary, public dataset and categorically commit that the code used for the analysis (e.g., deployment of the deep learning model) will be shared to ascertain full reproducibility. (The mention of SQL code on GitHub is a solid start, yet the code of the ML/DL model is critical).

4. The discussion acknowledges that the study has a limitation in evaluating if AF burden-directed interventions reduce outcomes. However, more reinforcing the potential mechanism of action (e.g., hemodynamic instability, inflammatory response) linking high AF burden to mortality would strengthen the argument. While the ≥7.0% threshold for "high AF burden" is borrowed from an earlier study, its clinical and biological logic in the setting of this manuscript may be clarified further in the introduction or discussion.

5. It is suggested that the authors need to mention about the technology foundation in the Introduction and Discussion, include a citation on the use of deep learning or AI in healthcare or IoT because it deserves the largest methodology. The following citation from the provided document is extremely relevant to the intersection of deep learning and health monitoring. The suggested reference paper from: “Deep Learning-Enabled Fetal Health Classification Through Sensor-Fused IoT. Mobile Radio Communications and 5G Networks”. This paper suggested explains where the lack of adequate research in AF burden and its clinical importance is expressed, to move to the solution using deep learning to the discussion to reinforce the present arguments in deep learning in healthcare and application of AI in risk stratification.

6. Although deep learning forms the crux of the method, the exposition is primarily based on the existing deep learning architecture (ResNet/SE-ResNet). A brief clarification regarding how this computerized high-frequency load calculation is specifically bridging the limitation of traditional manual or intermittent monitoring in a busy ICU scenario would be beneficial. Enhance Rationale for AF Burden Threshold: Provide a superior, evidence-based rationale in the discussion for employing the ≥7.0% threshold for high AF burden by explaining its clinical relevance or statistical derivation from prior work.

Reviewer #2: Dear Authors,

Thank you for the opportunity to review your manuscript entitled “Association Between Deep Learning–Based Atrial Fibrillation Burden and In-Hospital Mortality.” The study presents a novel and clinically relevant approach to quantifying atrial fibrillation (AF) burden using deep learning analysis of continuous ECG monitoring in critically ill patients. The integration of artificial intelligence with large-scale ICU datasets (MIMIC-III and an external Korean cohort) is impressive and has potential to enhance dynamic risk stratification in critical care settings.

However, several important issues need to be addressed before the paper can be considered for publication.

Methodological Transparency:

Please clarify how AF burden was computed and justify the chosen cutoff (≥7.0%) for defining “high burden.” Was this threshold determined empirically, or derived from prior literature or ROC analysis?

Provide more details on the deep learning model architecture (e.g., SE-ResNet-34), input parameters, and validation metrics (accuracy, sensitivity, specificity) used to ensure reliable AF detection.

Explain how ECG artifacts were managed and why patients with AF burden >0.9 were excluded.

Confounding and Adjustment:

Expand on the rationale for including variables such as SOFA score, sepsis, and ventilator use in the multivariable model. Discuss potential multicollinearity, and report VIFs if available.

Consider adjusting for additional confounders such as vasoactive drug use, baseline cardiac history, or renal replacement therapy.

Model Interpretability and Generalizability:

Provide quantitative data on feature importance (not only SHAP plots) to highlight the predictive contribution of AF burden relative to other variables.

In the external validation cohort, please describe ECG data collection parameters and confirm that preprocessing and classification pipelines were identical to the MIMIC-III cohort.

Clinical Implications:

The discussion could be strengthened by explaining how real-time AF burden monitoring might be integrated into ICU alert systems or decision support tools.

Minor Revisions:

Ensure consistent terminology for AF burden thresholds throughout the text.

Improve figure readability (particularly Fig. 2B and Fig. 3) and update references to include more recent AI–AF burden studies (post-2022).

In summary, this is a promising and methodologically sound study that could significantly contribute to the understanding of AF burden as a dynamic prognostic marker in ICU patients. With greater methodological transparency and clearer articulation of clinical applications, the paper will be substantially strengthened.

Best regards,

**Do you want your identity to be public for this peer review?** For information about this choice, including consent withdrawal, please see our Privacy Policy

Reviewer #1: No

Reviewer #2: **Yes:** Duaa Abualkhair, Division of Physiotherapy , Department of Applied and Allied Medical Sciences, Faculty of Medicine and Allied Medical Sciences, An-Najah National University, Nablus, Palestine . (www.najah.edu)

**Figure resubmission:**

**Reproducibility:**To enhance the reproducibility of your results, we recommend that authors of applicable studies deposit laboratory protocols in protocols.io, where a protocol can be assigned its own identifier (DOI) such that it can be cited independently in the future. Additionally, PLOS ONE offers an option to publish peer-reviewed clinical study protocols. Read more information on sharing protocols at https://plos.org/protocols?utm_medium=editorial-email&utm_source=authorletters&utm_campaign=protocols

---

## [Decision Letter · Decision Letter 1]

8 Feb 2026

Association Between Deep Learning–Based Atrial Fibrillation Burden and In-Hospital Mortality

PDIG-D-25-00824R1

Dear Dr. Yoon,

We are pleased to inform you that your manuscript 'Association Between Deep Learning–Based Atrial Fibrillation Burden and In-Hospital Mortality' has been provisionally accepted for publication in PLOS Digital Health.

Best regards,

Iqram Hussain, Ph.D.

Academic Editor

PLOS Digital Health

**Additional Editor Comments (if provided):**

**Reviewer Comments (if any, and for reference):**

Reviewer's Responses to Questions

**Comments to the Author**

Reviewer #1: All comments have been addressed

Reviewer #2: (No Response)

publication criteria?

Reviewer #1: Yes

Reviewer #2: Yes

3. Has the statistical analysis been performed appropriately and rigorously?

Reviewer #1: Yes

Reviewer #2: Yes

4. Have the authors made all data underlying the findings in their manuscript fully available (please refer to the Data Availability Statement at the start of the manuscript PDF file)?

Reviewer #1: Yes

Reviewer #2: Yes

5. Is the manuscript presented in an intelligible fashion and written in standard English?

Reviewer #1: Yes

Reviewer #2: Yes

Reviewer #1: I recommend accepting this manuscript. The authors have thoroughly addressed all the concerns. The updated manuscript significantly enhances the case for utilizing deep learning to assess Atrial Fibrillation (AF) burden in critical care settings.

1. The study goes beyond simply labeling AF as "present or absent." By defining AF burden as a continuous, changing measure, the authors give a clearer picture of how serious the arrhythmia is.

2. The authors applied an advanced SE-ResNet-34 model to analyze continuous ECG data from the ICU. This method effectively uses deep learning for automated analysis and overcomes the limitations of manual checks.

3. The authors have shown that AF burden is an important risk factor for in-hospital mortality (adjusted odds ratio: 1.63; 95% confidence interval: 1.36-1.95). They also demonstrate that this dynamic measure adds predictive value beyond traditional scores like SOFA.

4. The findings were confirmed in an independent group from Yongin Severance Hospital. The results were consistent across different datasets, such as MIMIC-III and the Korean cohort, highlighting the model's relevance in various settings.

5. The authors included a restricted cubic spline analysis, which effectively shows the significant and complex relationship between AF burden and mortality. This provides a stronger reason for understanding AF burden as a continuous measure.

6. The authors have made their deep learning model and analysis code available on GitHub, allowing others to reproduce their findings. This is essential for meaningful research in digital health.

Reviewer #2: The authors have adequately addressed all comments raised by the reviewers. The revisions have improved the clarity and quality of the manuscript, and I am satisfied with the responses provided.

I therefore recommend acceptance of the manuscript for publication.

**Do you want your identity to be public for this peer review?** For information about this choice, including consent withdrawal, please see our Privacy Policy

Reviewer #1: No

Reviewer #2: **Yes:** Duaa Abualkhair
